# DYN-ADAPTER: TOWARDS DISENTANGLED REPRESENTATION FOR EFFICIENT VISUAL RECOGNITION

## ABSTRACT

Parameter-efficient transfer learning (PETL) is a promising task, aiming to adapt the large-scale pretrained model to downstream tasks with a relatively modest cost. However, current PETL methods struggle in compressing computational complexity and bear heavy inference burden due to the complete forward process. This paper presents an efficient visual recognition paradigm, called *Dynamic Adapter (Dyn-Adapter)*, that boosts PETL efficiency by subtly disentangling features in multiple levels. Our approach is simple: first, we devise a dynamic architecture with balanced early heads for multi-level feature extraction, along with adaptive training strategy. Second, we introduce a bidirectional sparsity strategy driven by the pursuit of powerful generalization ability. These qualities enable us to fine-tune efficiently and effectively: we reduce FLOPs during inference by 50%, while maintaining or even yielding higher recognition accuracy. Extensive experiments on diverse datasets and pretrained backbones demonstrate the potential of *Dyn-Adapter* serving as a general efficiency booster for PETL. We will make the code publicly available.

## 1 INTRODUCTION

Very recently, large-scale deep neural networks have acheived remarkable advances and attracted growing interest in the vision community (Dosovitskiy et al., 2021; He et al., 2022; Radford et al., 2021; Tong et al., 2022; Zhai et al., 2022). These colossal models, often with billions of parameters, are pretrained on large datasets (e.g., ImageNet (Deng et al., 2009)) and then adapted to a multitude of downstream tasks (Lin et al., 2014; Goyal et al., 2017; Kuehne et al., 2011; Zhai et al., 2019; Zhou et al., 2019), demonstrating unprecedentedly strong capabilities. Such adaptation is usually done via fine-tuning in transfer learning, which typically updates all the parameters of the pre-trained model. However, with the rapidlly growing model size, directly fine-tuning these large-scale models can lead to prohibitively expensive storage overhead and computational cost(Luo et al., 2023; Chavan et al., 2023). To

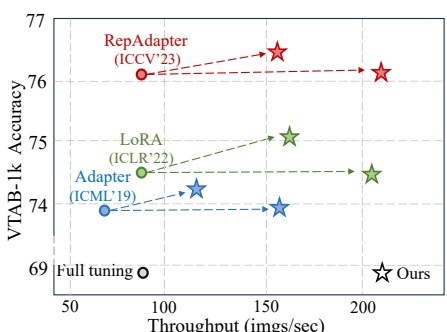

Figure 1: Performance comparison of our *Dyn-Adapter* and baselines. The throughput is measured on a NVIDIA 3090 GPU with a batch size of 1.

rectify this issue, research endeavours towards reducing the tuning cost using parameter-efficient transfer learning (PETL) methods (Hu et al., 2022; Chen et al., 2022; Jia et al., 2022; Zhang et al., 2022; Luo et al., 2023). PETL methods achieve efficient fine-tune by updating only a small number of parameters. By integrating light-weight modules or prepending additional learnable tokens to the input sequence, PETL methods can achieve comparable or even superior performance than full fine-tuning while keeping a significantly reduced parameter cost.

Despite the concerted efforts, existing PETL methods suffer from two drawbacks: **i)** inference efficency. Current literature cannot improve the inference efficiency of large-scale models, many methods even introduce additional architecture, resulting in extra latency and FLOPs overhead (Luo et al., 2023). Therefore, the parameter-efficient finetuning can not translate its theoretical advantages into practical efficiency. Since the application of PETL is usually resource-limited scenarios,

this drawback inevitably hinders its development; **ii)** entangled representation. According to *Information Bottleneck principle (IB)* (Tishby et al., 2000; Tishby & Zaslavsky, 2015), layers close to the input contain more low-level information, while features near the output are rich in semantic meanings. Although such learning paradigm achieves great success, it might not be the optimal choice in transfer learning: down-stream tasks may suffer from inferior performances if the learned features are over compressed, or the learned semantic information is irrelevant to the target tasks, especially if there exists a significant domain gap between the source and the target tasks (Cai et al., 2023). For PETL, distinct downstream datasets often possess unique characteristics, such as natural, specialized and structured data, which differ sharply in distribution and composition (Chavan et al., 2023). Nevertheless, current PETL methods are only capable of tuning the high-level semantic representations and can not directly utilize low-level information (e.g. locations of the edges) in case of the demand of down-stream tasks, thereby undermining their capacity to adapt to diverse datasets.

In this paper, we propose a novel PETL framework termed Dynamic Adapter (Dyn-Adpater). Specifically, we propose dynamic-balanced early heads to extract image features from a low level to a high level. These early heads directly act on the intermediate features of different layers and build connection with the task objective. For samples of different downstream tasks, our approach can dynamically decide which level of features to use depending on the input samples, which can not only improve the accuracy but also reduce unnecessary computation, hence boosting the inference efficiency. Notably, a critical problem in previous early-exiting iterature is that early classifiers force intermediate low-level features to encapsulate high-level semantics and be linearly separable, which destroys the inherent low-level feature in shallow layers and invariably backfire the performance (Huang et al., 2018; Han et al., 2023). In contrast, our approach overcomes this defect fundamentally. By freezing the backbone and only updating the adapter to assume the task-related semantic abstraction, we guarantee that the low-level feature in the backbone will not be interfered by loss signal, realizing explicit decoupling of low level feature and high level semantics. Furthermore, we introduce a bidirectional generalization strategy during the model's forward and backward propogation, which enhances the model's generalization ability and alleviates over-fitting.

Our framework boasts three essential advantages: **i)** fully explicit decoupling of feature extraction and early classification. The experiment results in Section 4.2 demonstrate that Dyn-Adapter prominently reduce inference latency with even superior performances; **ii)** the theoretical efficiency can effectively translate into practical speedup. Remarkably, our framework can eliminate 50% inference latency and FLOPs of PETL methods without backfiring performance, significantly enhancing their practical efficiency; **iii)** the simplicity and versatility of our framework. Our approach can be seamlessly migrated into existing PETL methods, consistently outperforming original methods with non-trivial margins.

To evaluate Dyn-Adapter, we apply it to multiple PETL methods including LoRA (Hu et al., 2022), Adapter (Houlsby et al., 2019) and RepAdapter (Luo et al., 2023) as Fig 1 shows. Extensive experiments across various vision tasks demonstrate our method's effectiveness. For instance, Our designs diminish RepAdapter's 50% inference latency and FLOPs without any compromise in accuracy on VTAB-1k (Zhai et al., 2019). Moreover, the visualization results exhibit that our method can preserve the low-level features of shallow layers, which further backups our motivation.

## 2   RELATED WORK

**Parameter-efficient Transfer Learning.** Parameter-efficient Transfer Learning (PETL) aims at fine-tuning a small number of trainable parameters to transfer large pre-trained models to downstream tasks. PETL was first introduced in the natural language processing (NLP) field (Houlsby et al., 2019; Hu et al., 2022; Lester et al., 2021; Li & Liang, 2021; Liu et al., 2023; Shin et al., 2020) and extended into large pre-trained vision models across a variety of vision tasks (Sung et al., 2022; Zhang et al., 2022; Zhou et al., 2022a;b; Chen et al., 2022; Lian et al., 2022; Luo et al., 2023). Generally, PETL methods integrate light-weight modules or prepend additional learnable tokens to the input sequence to adapt down-stream tasks while keeping the original backbone frozen. For instance, LoRA (Hu et al., 2022) proposes to freeze the pre-trained model weights and injects trainable low-rank decomposition matrices into each layer. VPT (Jia et al., 2022) proposes to insert a small number of learnable parameters as prompts and optimize them while freezing the backbone. SSF (Lian et al., 2022) module scales and shifts features after every MLP, MHSA, Layernorm mod-

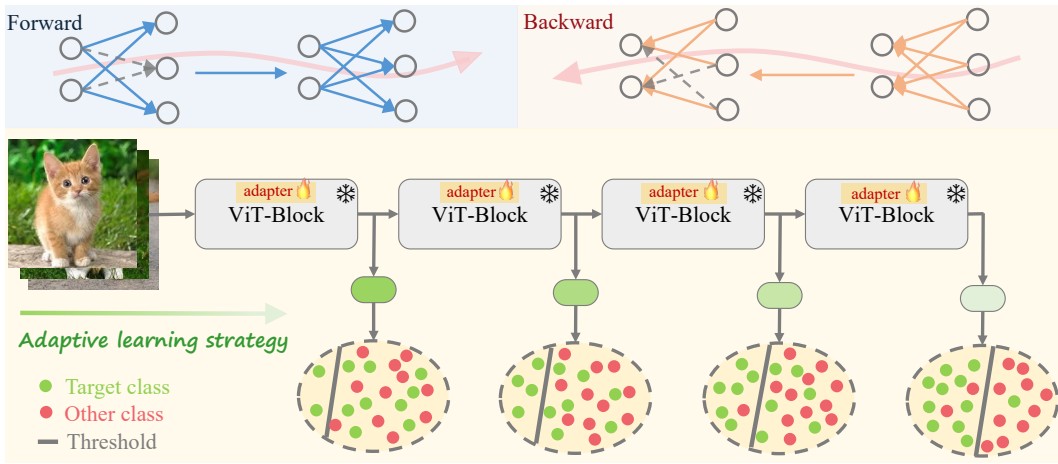

Figure 2: Overview of our *Dyn-Adapter* paradigm. Multiple early supervisions are introduced to facilitate dynamic inference (section 3.2). Adaptive learning and bidirectional sparsification strategy effectively addresses *Dyn-Adapter* optimization (section 3.3).

ule during training, and performs re-parameterization during inference as it is a linear structure. AdaptFormer (Chen et al., 2022) introduces a parallel learnable branch of two linear layers and ReLU over the MLP block and learns only this path while freezing other parts. RepAdapter (Luo et al., 2023) inserts sequential lightweight networks into both MHA and MLP, and the additional parameters will be re-parameterized to the nearby projection weights after training. In this paper, we propose a general framework which is applicable to all existing PETL methods. Without bells and whistles, our Dynamic Adapter can reduce PETL methods' FLOPs up to 50% without backfiring the fine-tuning accuracy, significantly improving the inference efficiency of PETL methods.

**Dynamic Early-exiting For Efficient Visual Recognition.** Dynamic networks (Bolukbasi et al., 2017; Graves, 2016; Figurnov et al., 2017; Huang et al., 2018; Yang et al., 2020) are designed to improve the inference efficiency of neural networks. Through adapting their computation commensurate with varying input complexities, dynamic networks have demonstrated promising results in efficient visual recognition (Han et al., 2021). For instance, Bolukbasi et al. (2017) allows examples correctly classified using early layers of the system to exit, and avoid the computational time associated with full evaluation of the network. RANet (Yang et al., 2020) proposes a resolution-based dynamic early-exiting framework, which processes simple samples with low resolution paths and hard samples with high resolution paths respectively. Despite these advances, a fatal problem exits: feature extraction and early classification are intricately intertwined as introduced in Section 1. As the ramification of this contradiction, classifiers are observed to interfere with each other and significantly degrade the performance of the final exit. To alleviate, Dynamic Perceiver (Han et al., 2023) proposes to decouple early classification and feature extraction with a two-branch structure and a latent code design. However, since the gradients can still be back-propagated to the shallow layers of the network, these designs can not realize complete decoupling of representations, and the low-level features of the shallow layers are still modified. Different from prior literatures, we start from a new perspective, freezing the backbone to keep the feature representation retained. In our Dynamic Adapter, only adapter is updated to abstract high-level semantic for classification, while the main backbone is frozen thus its low-level feature can be preserved. In this way, we realize fully-decoupling of feature extraction and classification, and the experimental results exhibit the great potential of dynamic early-exiting in the field of transfer learning.

## 3 METHODOLOGY

In this section, we introduce a simple yet elegant paradigm - *Dyn-Adapter*. In section 3.1, we first review the current PETL paradigm. In section 3.2, we introduce the overall framework of *Dyn-Adapter*, including approach setting, early head balance and adaptive priortization. Following this, in section 3.3 and 3.4, we present the bidirectional sparsification for more generalized gradient updates and dynamic inference process. The overall framework is illustrated in Figure 2.

## 3.1 PRELIMINARY

Below we briefly review two representative and top-performing PETL methods, *i.e.*, LoRA (Hu et al., 2022) and RepAdapter Luo et al. (2023).

**LoRA** freezes the pre-trained model weights and leverages trainable low-rank decomposition matrices layer in a parallel way. $\Delta W$ signifies the learnable low-rank decomposition weights. Presuming that $W_0$, $b_0$, and $X$ are the pre-trained weights, bias, and input respectively, and $g$ denotes a linear layer, then $g(X;\theta) = W_0 X + b_0$. The finetuning can be represented as follows and $\Delta W$ can be reparameterized during inference:

$$g(X, \theta) = W_0 X + \Delta W X + b_0 = W_{\text{LoRA}} X + b_0, \tag{1}$$

**RepAdapter** introduces a sequential adapter to both MHA and MLP. This adapter performs dense-to-sparse structure, where the upsampling projections is formulated as a group-wise transformation, further facilitate its lightweight characteristic. During inference, the adapter module can be structurally reparameterized and enjoys intact efficiency:

$$\begin{aligned}
g(X;\theta) &= W_0(X + W_u(W_d X + b_d) + b_u) + b_0 \\
&= (W_0 + W_0 W_u W_d)X + W_0 W_u b_d + W_0 b_u + b_0,
\end{aligned} \tag{2}$$

where $W_u$, $W_d$, $b_u$, and $b_d$ denote learnable weights and biases, respectively.

## 3.2 DYNAMIC ADAPTER

**Framework.** Motivated by the demand of high inference efficiency, *Dyn-Adapter* leverages early classification into the PETL methods. The incorporation of early classification allows for dynamic inference depth based on the complexity of the input. The overall framework is illustrated in Fig 2. Given an input image $I \in \mathbb{R}^{h \times w \times 3}$, ViT preprocesses it into a visual sequence $\boldsymbol{X}_0 \in \mathbb{R}^{n \times d}$, where n and d denote the token length and embedding dimension respectively. Then the visual sequence is fed into subsequent $N$ ViT blocks and supervision is performed with a stable interval $T$ by classification targets $Y$. The total number of introduced supervision $S$ can be easily calculated by $S = N/T$. Specifically, the early head and final head lies after the $l$-th ViT block, where $l = iT, i \in \{1, 2, \cdots, S\}$ and $i$ represents the index of supervision stage. The pre-trained ViT blocks are frozen and only the adapters are updated.

Taking the $l$-th block as example, given the output features of $(l\text{-}1)$-th block $\boldsymbol{X}_{l-1}$, the $i$-th ViT block employ computation and the $l$-th prediction $\hat{Y}_l$ is obtained as described in the following equation:

$$\begin{aligned}
\boldsymbol{X}_l' &= \text{MHA}(Adapter(LN(\boldsymbol{X}_{l-1}); \theta) + \boldsymbol{X}_{l-1}, \\
\boldsymbol{X}_l &= \text{FFN}(Adapter(LN(\boldsymbol{X}_l'); \theta)) + \boldsymbol{X}_l', \\
\hat{Y}_l &= \text{HEAD}(\boldsymbol{X}_l).
\end{aligned} \tag{3}$$

The objective is to minimize the classification loss between $\hat{Y}_l$ and the corresponding targets $Y$:

$$\mathcal{L} = \sum_{l=T}^{ST} \lambda_l \, \mathcal{L}_{cls}(\hat{Y}_l, Y), \tag{4}$$

where $\lambda_l$ is the weight of the classification supervision at the $l$-th block.

The core goal of *Dyn-Adapter* is to jointly optimize the early-exit target in the PEFL settings. For simplicity, we denote the feature extraction function (including frozen backbone and free adapter) of stage $i$ as $f_i$, and the classification head of stage $i$ as $c_i$. The prediction of stage $i$ can be represented as $\hat{Y}_{i+1} = c_{i+1} \circ f_{i+1}(\boldsymbol{X}_i)$. The following strategy addresses the *Dyn-Adapter* optimization by comprehensively considering the design of $\lambda$, $f$ and $c$.

**Head Balance.** Early classification heads $c$ play a crucial role in dynamic inference, while causing gradient interference as widely acknowledged in the supervised learning. When facilitated by the naturally collaborative characteristic of PETL and early supervision, we further observe that there exists inconsistency of optimization directions inducted by multiple supervisions. *What causes the collision?*

We study the correlation between different optimization directions and the design of $c$. Since $f$ and $c$ are sequentially arranged step by step, we choose the adjacent stages $i$ and $i + 1$ for analysis. The

prediction of adjacent stages can be obtained by $\hat{Y}_i = c_i(\boldsymbol{X}_i), \hat{Y}_{i+1} = c_{i+1} \circ f_{i+1}(\boldsymbol{X}_i)$ respectively. The following lemma depicts the inconsistency of optimization direction.

> **Lemma 1.** *The output of early exit heads shares the same optimization target, i.e.,*
> $$\hat{Y}_{i+1} \to Y, \hat{Y}_i \to Y.$$
>
> *However, in the past early exit scenarios, the heavy misalignment of path to obtain $\hat{Y}_i$ and $\hat{Y}_{i+1}$ may cause optimization direction interference intrinsically.*

Inspired by the Lemma 1, we propose the ideal requirement for $f$ and $c$ in Theorem!1.

> **Theorem 1.** *Consider the bond of feature representations among different blocks. Assuming that each part under the classification supervision has been optimized ideally, the relationship between the adjacent feature extraction block and the classification head can be expressed as:* $c_{i+1} \circ f_{i+1} \circ f_i(\boldsymbol{X}_{i-1}) = c_i \circ f_i(\boldsymbol{X}_{i-1})$, *which means*
> $$c_{i+1} \circ f_{i+1}(\cdot) = c_i(\cdot), i \in \{0, 1, 2, ..., S-1\}.$$

The ideal state of $f$ and $c$ can be represented as Eq 5, which performs chain structure.

$$c_1(\cdot) \sim c_2 \circ f_2(\cdot) \sim \cdots \sim c_S \circ f_S \circ f_{S-1} \circ \cdots \circ f_2(\cdot). \tag{5}$$

Note that $f_i$ is the ViT block function with considerable complexity, we alleviate the intrinsic interference by leveraging dynamic head in a hierarchical manner. Specifically, we employ classification in the early stages with heavier heads (i.e., MLP layers), while late stages possess light-weighted heads for decision. Such design allocates more burden for heads hanging after shallow layers, endowing the network with stronger potential for joint optimization, which is enlightened by the theoretical perspective.

**Adaptive Prioritization.** We introduce adaptive weight prioritization for multi-stage learning. Supervision of different stages play various roles in the joint learning process: **i)** due to the sufficient semantic features borrowed from pre-trained model in the deep layers, the late classification heads do better in hard sample classification, while shallow layers prefer easier ones. **ii)** The insertion of early supervision primarily aims to improve inference efficiency, and the upper bound of recognition is still determined by the deep layers. **iii)** The optimization guided by the late supervision may influence both shallow and deep gradient update, which implies it plays a more critical function for general optimization direction.

Therefore, it is necessary to adaptively adjust the prioritization of $\lambda$ according to the analysis. The design of $\lambda$ follows the guidelines respectively: **i)** for harder objectives, *i.e.*, late classification, their prioritization need to be preposed for better learning potential and possible interference avoidance. **ii)** No matter in what learning period, the weight of layers handling the recognition upper bound should be guaranteed and preserved. **iii)** More generalized layers can lay the foundation for the subsequent specific task, such as early classification.

Guided by the policy, we subtly design the prioritization of $\lambda$. At the beginning period, the weight of the deep supervision $\lambda_{deep}$ should be initialized as a relative large value to ensure the classification ability of the deep layers, and the weight of shallow layers $\lambda_{shallow}$ is set to a relatively small value to protect the learning process of deeper layers. Subsequently, $\lambda_{shallow}$ can be progressively increased, and $\lambda_{deep}$ gradually declines. The upper bound of $\lambda_{shallow}$ equals to the lower bound of $\lambda_{deep}$ to the extent that the hard sample classification ability is preserved, achieving dynamic inference based on data difficulty while ensuring a strong classification ability.

### 3.3 BIDIRECTIONAL SPARSIFICATION STRATEGY

Through the design of the overall framework and training strategy, we inspire *Dyn-Adapter* to arrange $\lambda$ and $c$ according to its intrinsic nature, facilitating optimization of dynamic paradigm. For the sake of comprehensive design, we further investigate the feature extraction module in *Dyn-Adapter* - $f$. When large pre-trained models are used for downstream fine-tuning, overfitting easily occurs, hence enhancing the generalization ability is crucial. In *Dyn-Adapter*, the deep blocks are used for deep feature extraction only, while the shallow features are used for late classification and early decision both, then Theorem 2 comes.

**Theorem 2.** *The features extracted from shallow layers are reused by multiple supervisions, thus assuming more functions, while features from deep layers are more free to be supervised by deep classification only. Considering this characteristic, the generalization ability of shallow layers should be enhanced.*

Inspired by Theorem 2, we think about both forward and backward propagation processes deeply and employ bidirectional sparsification strategy to strengthen the generalization power and robustness.

**Forward Process.** During the forward process, task collaboration of shallow blocks may cause suboptimal performance. $p_{shallow}$ and $p_{deep}$ denote the dropout probability of shallow layers and deep layers respectively. We set $p_{shallow} > p_{deep}$, implementing dynamic dropout. A more drastic dropout $p_{shallow}$ naturally alleviates the issues caused by collaborative effects. By noticeably dropping some nodes, the nodes in the network acquire relatively task-agnostic capability. $p_{deep}$ is set to a normal value, allowing more nodes to focus on learning high-level classifications. This setting allows for a more flexible and adaptive network and enjoys several charms: **i)** multi-path forward combination brought by sparsification implies a voting mechanism, which contributes to more robust features. **ii)** Sparsification encourages the nodes to learn towards objectives independently, eliminating joint adaptability between neuron nodes and enhancing the generalization capability.

**Backward Process.** During the conventional back-propagation process, the weights $\mathbf{w}$ of all parameters are updated, which results in a relatively fixed paradigm. To endow the gradient updates with a larger combinatorial capacity and relieve overfitting, we employ a masked gradient update strategy.

Specifically, we randomly generate a gradient mask $M$ with a certain mask probability $p_m$. In the backward process, the gradients corresponding to a mask value of 0 are not updated, while those with a mask value of 1 are updated. Given that the mask is randomly generated each time, there will be a diverse combinations for gradient updates, allowing for more flexible backward path and stronger generalization capabilities, which can be mathematically expressed with:

$$\Delta\mathbf{w} = \alpha\frac{\partial\mathcal{L}(\mathbf{w})}{\partial\mathbf{w}} \odot M, \tag{6}$$

where $\alpha$ represents the coefficient of gradient update, and $\odot$ is the dot product operation.

### 3.4 DYNAMIC INFERENCE

During inference, we dynamically adjust the network depth based on the complexity level of input samples. As shown in Fig 3, when the early stage struggles in handling the input sample and the confidence (the max value of the *softmax* probability) fails to reach the threshold, the network will step into the subsequent stage. Once the confidence exceeds the threshold, the inference process exits. The final classification results are obtained depending on the input characteristic, which is consistent with our design in training process.

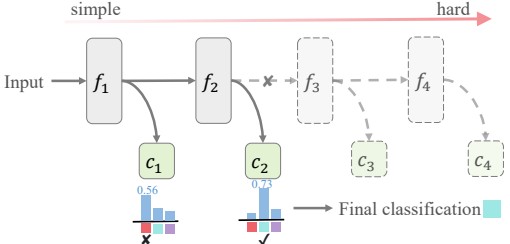

Figure 3: Dynamic inference process.

## 4 EXPERIMENT

### 4.1 EXPERIMENT SETUP

**Datasets and Metrics.** We leverage VTAB-1k (Zhai et al., 2019) benchmark to evaluate the transfer learning performance of our approach. VTAB-1k contains 19 dataset subsets, which can be grouped into *Natural*, *Specified* and *Structured* categories. Each subset contains 1000 labeled images, in which 800 images are split into `train` and 200 images are for `val`. When inducting few-shot learning experiment, five fine-grained datasets (*i.e.*, Food-101, StanfordCars, Flowers102, FGVCAircraft and OxfordPets) are leveraged. For the domain generalization, we train the model on ImageNet and test on four other variants of ImageNet (*i.e.*, ImageNetV2, ImageNet-Sketch, ImageNet-A and ImageNet-R) that perform various types of domain shift. We employ *Top-1 classification accuracy* as metric.

Table 1: Results on VTAB-1k benchmark. ViT-B/16 pretrained on ImageNet-21k is used as the vision model of all methods. Our framework can reduce PETL methods' FLOPs by 50% without backfiring performance, or achieve a noticeable improvement with a 30% reduction in FLOPs.

| Method | Param (M) | FLOPs (G) | Throughput (imgs/s) | Avg. Acc. | Cifar100 | Caltech101 | DTD | Flowers102 | Pets | SVHN | Sun397 | Camelyon | EuroSAT | Resist45 | Retinopathy | Clevr-Count | Clevr-Dist | DMLab | KITTI-Dist | dSpr-Loc | dSpr-Ori | sNORB-Azim | sNORB-Ele |
|---|---|---|---|---|---|---|---|---|---|---|---|---|---|---|---|---|---|---|---|---|---|---|---|
| | | | | | | | Natural | | | | | Specialized | | | | Structured | | | | | | | |
| *Conventional FT* | | | | | | | | | | | | | | | | | | | | | | | |
| Full tuning | 85.8 | 16.8 | 91.3 | 68.9 | 68.9 | 87.7 | 64.3 | 97.2 | 86.9 | 87.4 | 38.8 | 79.7 | 95.7 | 84.2 | 73.9 | 56.3 | 58.6 | 41.7 | 65.5 | 57.5 | 46.7 | 25.7 | 29.1 |
| Linear probe | 0.04 | 16.8 | 90.6 | 57.6 | 64.4 | 85.0 | 63.2 | 97.0 | 86.3 | 36.6 | 51.0 | 78.5 | 87.5 | 68.5 | 74.0 | 34.3 | 30.6 | 33.2 | 55.4 | 12.5 | 20.0 | 9.6 | 19.2 |
| *PETL methods* | | | | | | | | | | | | | | | | | | | | | | | |
| VPT | 0.53 | 22.4 | 87.7 | 72.0 | 78.8 | 90.8 | 65.8 | 98.0 | 88.3 | 78.1 | 49.6 | 81.8 | 96.1 | 83.4 | 68.4 | 68.5 | 60.0 | 46.5 | 72.8 | 73.6 | 47.9 | 32.9 | 37.8 |
| AdaptFormer | 0.16 | 16.9 | 70.4 | 74.7 | 70.8 | 91.2 | 70.5 | 99.1 | 90.9 | 86.6 | 54.8 | 83.0 | 95.8 | 84.4 | 76.3 | 81.9 | 64.3 | 49.3 | 80.3 | 76.3 | 45.7 | 31.7 | 41.1 |
| NOAH | 0.36 | 16.9 | 72.3 | 75.5 | 69.6 | 92.7 | 70.2 | 99.1 | 90.4 | 86.1 | 53.7 | 84.4 | 95.4 | 83.9 | 75.8 | 82.8 | 68.9 | 49.9 | 81.7 | 81.8 | 48.3 | 32.8 | 44.2 |
| SSF | 0.24 | 16.8 | 90.3 | 75.7 | 69.0 | 92.6 | 75.1 | 99.4 | 91.8 | 90.2 | 52.9 | 87.4 | 95.9 | 87.4 | 75.5 | 75.9 | 62.3 | 53.3 | 80.6 | 77.3 | 54.9 | 29.5 | 37.9 |
| Adapter | 0.16 | 16.9 | 69.9 | 73.9 | 69.2 | 90.1 | 68.0 | 99.8 | 89.9 | 82.8 | 54.3 | 84.0 | 94.9 | 81.9 | 75.5 | 80.9 | 65.3 | 48.6 | 78.3 | 74.8 | 48.5 | 29.9 | 41.6 |
| Dyn-Adapter | +0.12 | **8.5** | **154.7** | **73.9** | 68.3 | 90.8 | 68.4 | 98.9 | 88.7 | 84.7 | 54.3 | 83.5 | 95.8 | 84.7 | 75.8 | 78.9 | 64.6 | 47.4 | 78.3 | 74.2 | 47.1 | 29.6 | 39.6 |
| | +0.12 | **11.7** | **116.6** | **74.2** | 68.3 | 91.1 | 67.6 | 98.9 | 89.5 | 85.7 | 54.3 | 83.0 | 95.8 | 84.6 | 75.8 | 80.4 | 64.8 | 48.5 | 78.5 | 76.0 | 49.4 | 30.0 | 40.0 |
| LoRA | 0.29 | 16.8 | 90.5 | 74.5 | 67.1 | 91.4 | 69.4 | 98.8 | 90.4 | 85.3 | 54.0 | 84.9 | 95.3 | 84.4 | 73.6 | 82.9 | 69.2 | 49.8 | 78.5 | 75.7 | 47.1 | 31.0 | 44.0 |
| Dyn-LoRA | +0.17 | **8.5** | **204.2** | **74.5** | 67.7 | 90.5 | 70.0 | 99.0 | 89.4 | 86.3 | 53.6 | 86.2 | 95.7 | 84.3 | 75.0 | 79.9 | 67.3 | 48.5 | 81.9 | 77.8 | 45.4 | 31.2 | 38.4 |
| | +0.17 | **11.7** | **159.1** | **75.0** | 67.9 | 90.5 | 70.4 | 99.1 | 89.8 | 86.4 | 53.6 | 86.3 | 95.7 | 84.3 | 75.1 | 81.6 | 67.8 | 50.2 | 82.1 | 79.1 | 47.0 | 31.6 | 39.6 |
| Repadapter | 0.22 | 16.8 | 90.9 | 76.1 | 72.4 | 91.6 | 71.0 | 99.2 | 91.4 | 90.7 | 55.1 | 85.3 | 95.9 | 84.6 | 75.9 | 82.3 | 68.0 | 50.4 | 79.9 | 80.4 | 49.2 | 38.6 | 41.0 |
| Dyn-Repadapter | +0.16 | **8.5** | **207.5** | **76.1** | 71.9 | 92.2 | 71.2 | 99.2 | 89.9 | 90.4 | 54.3 | 85.7 | 96.1 | 86.3 | 76.1 | 78.7 | 68.2 | 49.8 | 81.0 | 82.4 | 48.5 | 36.4 | 41.9 |
| | +0.16 | **11.7** | **156.7** | **76.4** | 71.8 | 92.6 | 71.7 | 99.1 | 90.6 | 90.8 | 54.3 | 85.8 | 95.9 | 86.4 | 76.1 | 80.3 | 68.9 | 49.9 | 81.9 | 82.3 | 50.3 | 36.8 | 41.1 |

Table 2: Efficiency comparison of our method and existing PETL methods during inference. Our theoretical efficiency can effectively translate into practical speedup.

| Method | $\Delta F$ (G) | GPU latency (imgs/sec) | | | | |
|---|---|---|---|---|---|---|
| | | bs=1 | bs=4 | bs=16 | bs=64 | bs=128 |
| Full tuning | 0 | 91.5 | 375.7 | 539.5 | 568.8 | 578.3 |
| VPT | +5.60 | 86.1 (-5.9%) | 283.5 (-24.5%) | 381.5 (-29.2%) | 406.7(-28.5%) | 421.6 (-27.1%) |
| Adapter | +0.03 | 70.9 (-22.5%) | 306.6 (-18.3%) | 504.7 (-6.4%) | 533.5 (-6.2%) | 552.4 (-5.8%) |
| AdapterFormer | +0.03 | 71.4 (-21.9%) | 309.9 (-17.5%) | 508.1 (-4.2%) | 546.0 (-4.0%) | 555.2 (-3.9%) |
| NOAH (500ep) | +0.02 | 72.1 (-21.2%) | 312.7 (-16.7%) | 492.9 (-8.6%) | 523.9 (-7.9%) | 534.7 (-7.5%) |
| Repadapter | 0 | 91.5 (-0.0%) | 375.7 (-0.0%) | 539.5 (-0.0%) | 568.8(-0.0%) | 578.3 (-0.0%) |
| Dyn-Repadapter | -8.30 | 202.7 (+121.5%) | 843.3 (+124.4%) | 1228.7 (+127.7%) | 1338.9 (+135.4%) | 1369.8 (+136.9%) |

**Implementation Details.** We employ ViT-Base (ViT/16) (Dosovitskiy et al., 2021) pre-trained on ImageNet-21k (Deng et al., 2009) with supervision as default backbone. We empirically set the supervision stages $S = 4$ and insert supervisions uniformly. The upper bound of $\lambda_{shallow}$ $\sup(\lambda_{shallow})$ equals to $\lambda_{deep}$ $\inf(\lambda_{deep})$ which is 0.5. The hyper-parameter $p_{shallow}$ and $p_{deep}$ are 0.5 and 0.1 respectively, and gradient mask probability $p_m = 0.1$. For all models, we trained for 100 epochs. The throughput and GPU latency in this paper are all tested on NVIDIA 3090 GPU. Other details including data augmentation and initialization are consistent with previous work (Hu et al., 2022; Luo et al., 2023).

Table 3: Image classification accuracy for SSL pretrained objectives. Our method is also well suited to contrastive learning (e.g., MoCo-v3) and masked image modeling (e.g., MAE) methods.

| Pretrained objectives | MoCo v3 | | | | | | MAE | | | | | |
|---|---|---|---|---|---|---|---|---|---|---|---|---|
| Method | P (M) | F (G) | Acc. | VTAB-1k | | | P (M) | F (G) | Acc. | VTAB-1k | | |
| | | | | *Natural* | *Specialized* | *Structured* | | | | *Natural* | *Specialized* | *Structured* |
| Full tuning | 85.8 | 16.8 | 69.55 | 71.95 | 84.72 | 51.98 | 85.8 | 16.8 | 64.27 | 59.31 | 79.68 | 53.82 |
| Linear probe | 0.04 | 16.8 | 59.62 | 67.46 | 81.08 | 30.33 | 0.04 | 16.8 | 32.10 | 18.87 | 53.72 | 23.70 |
| VPT | 0.53 | 22.4 | 65.23 | 70.27 | 83.04 | 42.38 | 0.53 | 22.4 | 41.07 | 36.02 | 60.61 | 26.57 |
| Adapter | 0.16 | 16.9 | 68.18 | 74.19 | 82.66 | 47.69 | 0.16 | 16.9 | 56.36 | 54.90 | 75.19 | 38.98 |
| Lora | 0.29 | 16.8 | 70.84 | 69.84 | 83.71 | 58.98 | 0.29 | 16.8 | 70.28 | 65.99 | 82.61 | 62.23 |
| Dyn-LoRA | +0.17 | **8.5** | **72.33** | 73.51 | 85.32 | 58.16 | +0.17 | **8.5** | **68.30** | 66.11 | 82.94 | 55.84 |
| | +0.17 | **11.7** | **73.07** | 73.81 | 85.48 | 59.92 | +0.17 | **11.7** | **70.36** | 66.53 | 84.13 | 60.42 |
| Repadapter | 0.22 | 16.8 | 72.03 | 71.82 | 84.27 | 60.01 | 0.22 | 16.8 | 69.46 | 66.15 | 81.89 | 60.35 |
| Dyn-Repadapter | +0.16 | **8.5** | **72.11** | 73.53 | 85.57 | 57.22 | +0.16 | **8.5** | **68.37** | 65.54 | 82.90 | 56.67 |
| | +0.16 | **11.7** | **73.49** | 74.69 | 85.83 | 59.96 | +0.16 | **11.7** | **70.45** | 67.05 | 83.71 | 60.60 |

Table 4: Results of 16-shot image classification on few-shot learning datasets.

| Method | Param (M) | FLOPs (G) | Avg. Acc. | Food-101 | StanfordCars | Flowers102 | FGVCAircraft | OxfordPets |
|---|---|---|---|---|---|---|---|---|
| VPT | 0.13 | 22.4 | 72.0 | 72.6 | 56.0 | 99.4 | 42.5 | 89.6 |
| Adapter | 0.24 | 16.9 | 73.2 | 71.7 | 60.4 | 99.5 | 45.2 | 89.1 |
| LoRA | 0.38 | 16.8 | 75.3 | 72.5 | 68.2 | 99.6 | 47.6 | 88.7 |
| NOAH (500ep) | 6.69 | 16.9 | 76.5 | 76.3 | 68.6 | 99.5 | 49.1 | 89.0 |
| Repadapter | 0.43 | 16.8 | 74.9 | 74.6 | 65.7 | 99.4 | 44.8 | 89.8 |
| Dyn-Repadapter | +0.05 | 11.6 | 74.9 | 73.3 | 66.6 | 99.6 | 45.6 | 89.3 |

## 4.2 EXPERIMENTAL RESULTS

### 4.2.1 COMPARISON TO STATE-OF-THE-ARTS

We employ proposed *Dyn-Adapter* paradigm on three classic baseline methods including Adapter (Houlsby et al., 2019), LoRA (Hu et al., 2022) and RepAdapter (Luo et al., 2023). As shown in Table 1, our paradigm stably boost inference efficiency and preserve base accuracy without any compromise. *Dyn-Adapter* maintain or slightly outperform baseline methods in the dramatic 50% FLOPs decline case. When the inference FLOPs approximately equal to 70% of corresponding baseline, the accuracy are further yielded to a higher level (*i.e.*, +0.3% to +0.5%). Notably, Dyn-RepAdapter set new *state-of-the-art*, surpassing the baseline with 0.3% accuracy and save 30% computational complexity simultaneously, demonstrating strong adapting ability of *Dyn-Adapter*.

### 4.2.2 EFFICIENCY ANALYSIS

Inference speed lies in a crucial position in PETL performance analysis. Table 2 lists FLOPs variation ($\Delta F$) and GPU latency tested on NVIDIA 3090 GPU of several PETL methods. Traditional PETL methods bring increase in computational complexity due to the inserted module, which causes latency of varying degrees. LoRA and RepApater smartly design the adapter module and its inserted position to implementing re-parameter strategy during inference, leading to zero FLOPs change and latency. They have gained a significant advantage for this attribute. When it comes to *Dyn-Adapter*, benefiting from dynamic inference based on input, it has achieved a sharp reduction in computational load and latency for the first time, making a new breakthrough in improving reasoning efficiency.

### 4.2.3 GENERALIZATION EXPERIMENTS

**More Pre-trained Objectives.** We explore the performance of *Dyn-Adapter* with SSL pretrained objectives, *i.e.*, Moco v3 (Chen et al., 2021) and MAE (He et al., 2022), which are representative works for contrastive learning and masked image modeling respectively. The performance on MAE is consistently stable, and the results on Moco v3

Table 5: Results in domain generalization.

| Method | P (M) | F (G) | Source | Target | | |
|---|---|---|---|---|---|---|
| | | | ImageNet | -V2 | -Sketch | -A | -R |
| VPT | 0.82 | 22.4 | 70.5 | 58.0 | 16.4 | 4.6 | 23.2 |
| Adapter | 0.93 | 16.9 | 70.5 | 59.1 | 16.4 | 5.5 | 22.1 |
| NOAH (500ep) | 7.38 | 16.9 | 71.7 | 66.1 | 24.8 | 11.9 | 28.5 |
| LoRA | 1.06 | 16.8 | 70.8 | 59.3 | 20.0 | 6.9 | 23.3 |
| Dyn-LoRA | +2.21 | 12.4 | 71.0 | 59.3 | 20.7 | 7.3 | 22.5 |

are even more outstanding. Under the condition of 50% FLOPS, our method can boost accuracy by 1.5% based on LoRA, which signifies a perfect combination of extreme inference speed and notable performance enhancement.

**Few-shot Learning.** Following NOAH (Zhang et al., 2022), we conduct 16-shot few-shot learning on five FGVC datasets as Table 4. Reducing about 30% FLOPs, our approach exhibits comparable performance to the baseline under few-shot condition, demonstrating the robust ability of our method to transfer based on a few samples.

**Domain Generalization.** The capacity of out-of-domain generalization becomes crucial criterion for measuring PETL methods. Fine-tuning on ImageNet with custom 16-shot setting, we evaluate domain generalization ability by directly adapt to four variants of ImageNet with severe domain

shift. As implied by Table 5, our approach maintains a comparable performance in the hard domain shift case, stably occupying 75% computation.

### 4.2.4 ABLATION STUDIES

| Table 6: Componenent. | | Table 7: Heavier head position. | | | Table 8: Priorty setting. | | | Table 9: Dropout rate. | |
|---|---|---|---|---|---|---|---|---|---|
| Setting | Acc. | Shallow | Deep | Acc. | Shallow | Deep | Acc. | Rate | Acc. |
| Baseline | 73.6 | - | - | 75.8 | - | - | 75.0 | 0.1 | 75.7 |
| + Priorty | 75.5 | ✓ | - | 76.4 | ✓ | - | 74.3 | 0.3 | 76.1 |
| + Head Bal. | 74.5 | - | ✓ | 76.0 | - | ✓ | 76.4 | 0.5 | 76.4 |
| + Sparsity. | 74.7 | ✓ | ✓ | 76.1 | ✓ | ✓ | 75.4 | 0.7 | 75.2 |

We conduct ablation studies based on RepAdapter, with FLOPS controlled about 70% of baseline.

**Component Analysis.** We validate different components of Dyn-Repadapter in Table 6. As shown, adaptive prioritization strategy boosts performance with a large margin, which grasps the core of *Dyn-Adapter* optimiazation. The paradigm also benefits from bidirectional sparsification and head balance apparently.

**Head Capacity.** We explore the impact of the weights of shallow and deep classification heads. In Table 7 ✓indicates heavier heads. When the heads hanging on the shallow layers are heavier, the best performance is achieved, which further strengthens our reasoning.

**Learning Priority.** Even with detailed theoretical reasoning, we also experimentally prove the necessity of priority setting in Table 8. In the setting that shallow layers and deep layers are both marked ✓, they perform cross-optimization art. When the shallow and deep layers are optimized equally, the gradients interfere with each other. When the shallow layers are optimized first, the resulting bias makes it difficult for the deep features to learn, limiting the performance ceiling, causing sub-optimal accuracy. In contrast, our adaptive priority is the best choice.

**Dynamic Dropout.** For the drop rate of shallow layers, we employ various dropout rate in Table 9 and finally find that a large dropout rate – 0.5 provides shallow layer more generalization potential.

### 4.2.5 FEATURE VISUALIZATION AND ANALYSIS

**Disentangled Characteristic.** We visualize the CKA similarity (Kornblith et al., 2019) of output of ViT block/intermediate features of Dyn-Perceiver (Han et al., 2023) from shallow to deep level and label (normalized to $[0, 1]$). As shown in Fig 4, the previous early exit method (*i.e.*, Dyn-Perceriver) inevitably introduces supervisory information into the shallow

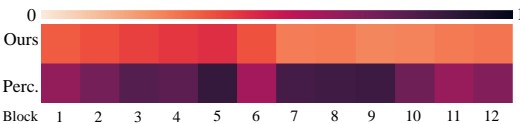

Figure 4: CKA of corresponding features and labels. Perc. is the abbreviation of Dyn-Perceiver.

layers, disrupting the learning paradigm of low-level features in shallow layers and semantic features in higher layers. However, in *Dyn-Adapter*, the adapter module bears the burden of aggregating high-level semantics, while freezing backbone maintains the information from large-scale pre-training, revealing the disentangled characteristic and further facilitating the optimization of *Dyn-Adapter*.

## 5 CONCLUSION

This paper proposes a novel and effective PETL paradigm, *Dyn-Adapter*. Inspired by the natural conflict-free characteristic of PETL and early supervision, we take the leading in exploring PETL with dynamic inference function, which explicitly decouple feature extraction and early classification and greatly boosts the inference efficiency without accuracy compromise. Based on the fresh framework, we subtly design the core component of *Dyn-Adapter* – early head balance, multi-stage weight prioritization and more generalized feature extraction, comprehensively addressing the adaptive optimization of *Dyn-Adapter*. Our efforts provides a deep insight about promoting inference computation without accuracy decline, which shed light on efficient and effective PETL paradigm.

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
