# SUPPLEMENTARY MATERIAL OF DYN-ADAPTER

## A DATASET

We have conducted experiments based on a large and diverse set of datasets. All the datasets leveraged are listed in Table 1.

Table 1: Datasets we leveraged for multiple experiments.

| | Dataset | Classes | Train | Val | Test |
|---|---|---|---|---|---|
| | VTAB-1k (Zhai et al., 2019) | | | | |
| *Natural* | CIFAR 100 (Krizhevsky et al., 2009) | 100 | | | 10000 |
| | Caltech101 (Fei-Fei et al., 2004) | 102 | | | 6084 |
| | DTD (Cimpoi et al., 2014) | 47 | | | 1880 |
| | Oxford-Flowers 101 (Nilsback & Zisserman, 2006) | 102 | 800/1000 | 200 | 6149 |
| | Oxford-Pets (Parkhi et al., 2012) | 37 | | | 3669 |
| | SVHN (Netzer et al., 2011) | 10 | | | 26032 |
| | Sun397 (Xiao et al., 2010) | 397 | | | 21750 |
| *Specialized* | Patch Camelyon | 2 | | | 32768 |
| | EuroSAT (Helber et al., 2019) | 10 | 800/1000 | 200 | 5400 |
| | Resisc45 (Cheng et al., 2017) | 45 | | | 6300 |
| | Retinopathy (Graham, 2015) | 5 | | | 42670 |
| *Structured* | Clevr/count (Johnson et al., 2017) | 8 | | | 15000 |
| | Clevr/distance Johnson et al. (2017) | 6 | | | 15000 |
| | DMLab (Beattie et al., 2016) | 6 | | | 22735 |
| | KITTI-Dist (Geiger et al., 2013) | 4 | 800/1000 | 200 | 711 |
| | dSprites/location (Matthey et al., 2017) | 16 | | | 73728 |
| | dSprites/orientation (Matthey et al., 2017) | 16 | | | 73728 |
| | SmallNORB/azimuth (LeCun et al., 2004) | 18 | | | 12150 |
| | SmallNORB/elevation (LeCun et al., 2004) | 18 | | | 12150 |
| | Few-shot Learning | | | | |
| | Food-101 (Bossard et al., 2014) | 101 | | 20200 | 30300 |
| | Stanford Cars (Krause et al., 2013) | 196 | | 1635 | 8041 |
| | Oxford-Flowers (Nilsback & Zisserman, 2006) | 102 | 16 per class | 1633 | 2463 |
| | FGVC-Aircraft (Maji et al., 2013) | 100 | | 3333 | 3333 |
| | Oxford-Pets (Parkhi et al., 2012) | 37 | | 736 | 3669 |
| | Domain Generalization | | | | |
| | ImageNet (Deng et al., 2009) | 1000 | 16 per class | 50000 | 50000 |
| | ImageNetv2 (Recht et al., 2019) | 1000 | - | - | 10000 |
| | ImageNet-Sketch (Wang et al., 2019) | 1000 | - | - | 50000 |
| | ImageNet-A (Hendrycks et al., 2021b) | 200 | - | - | 7500 |
| | ImageNet-R (Hendrycks et al., 2021a) | 200 | - | - | 30000 |

## B DISCUSSION

### B.1 LIMITATION

Although we think deeply about the optimization process of *Dyn-Adapter* from a theoretical perspective, conducting extensive experiments and achieving inspiring results, there still lacks profound

mathematical modeling for this joint optimization problem to elucidate from a more fundamental standpoint which direction is more optimal. Moreover, while the addition of an early head significantly enhances inference efficiency and ensures accuracy, the increment of the head inevitably introduces a minor portion of training parameters. Contemplating more efficient learning strategies, such as employing parameter sharing strategies to allow them to share some knowledge, is a direction that worth thinking in the future. We expect that future improvements based on our concise and efficient method will yield more desirable benefits.

### B.2 BROADER IMPACT

This work can benefit the wide application scenarios of PETL methods, and further reduce the inference efficiency with a large gap. Under resource-limited circumstances, our method provide more possibilities for the widespread utilization of PETL and save hardware resources in practice.

### B.3 FUTURE WORK

The aforementioned limitations demonstrate the directions of our future work. Moreover, investigating exit decision-making process in *Dyn-Adapter* is meaningful, as the right decision precedes the efficient early exit. Currently, the threshold of early stopping relies on thresholds induced from the training set, without explicit modeling of the network's inherent attributes, features of the pre-trained model, and transferred information within the adapter. There may exists several secrets to explore.