# OpenReview forum: "Dyn-Adapter: Towards Disentangled Representation for Efficient Visual Recognition"
_ICLR.cc/2024/Conference — ICLR 2024 Conference Withdrawn Submission_

### Official Review · Reviewer_6BHQ · 2023-10-30

**Soundness:** 2 fair
**Presentation:** 2 fair
**Contribution:** 2 fair
**Rating:** 5
**Confidence:** 5

**Summary:**

The paper introduces a new visual recognition paradigm called Dynamic Adapter (Dyn-Adapter) that focuses on improving the efficiency of parameter-efficient transfer learning (PETL). The approach involves disentangling features in multiple levels and implementing a bidirectional sparsity strategy. The experiments conducted show that Dyn-Adapter can effectively reduce computational complexity while maintaining or improving recognition accuracy. The authors also plan to make the code publicly available.

**Strengths:**

+ The motivation is clear and reasonable. Efficiency is indeed an important topic with current  PETL methods such as adapter.

+ The experiemnts are convincing.

+ The writing is easy to read and the topic is acceptable for ICLR.

**Weaknesses:**

- Method is not novel. Dynamic networks[1] and multi-level surpervision have been widely used in computer vision, Author should discuss the differences between this method and existing dynamic networks.


[1]. Han Y, Huang G, Song S, et al. Dynamic neural networks: A survey[J]. IEEE Transactions on Pattern Analysis and Machine Intelligence, 2021, 44(11): 7436-7456.

**Questions:**

I think the paper does solve some problems. If it can solve the novelty question, I am inclined to accept the paper.

---

### Official Review · Reviewer_jPC2 · 2023-10-31

**Soundness:** 2 fair
**Presentation:** 2 fair
**Contribution:** 2 fair
**Rating:** 3
**Confidence:** 4

**Summary:**

This paper tackles the inference efficiency and entangled representation aspects of parameter-efficient transfer learning  (PETL) . It proposes Dynamic Adapter (Dyn-Adapter) to add prediction heads for intermediate features. During inference, if the confidence score of a prediction head is less than a threshold, the features will continue to be processed by later layers, if the confidence exceeds the threshold, the inference process exits. It also proposes to use a higher dropout rate for shallow layers and a random masked gradient update for better generalization.

**Strengths:**

- The paper tries to address important problems in PETL
- The proposed method improves the throughput by a large margin while maintaining accuracy performance.

**Weaknesses:**

- Considerable improvements are needed in the writing and presentation quality.
    - Figure 2 should show what the dashed lines mean
    - "supervision is performed with a stable interval T by classification targets Y" and "The total number of introduced supervision S". It's quite confusing. I can only get the idea when I read the later parts of the paper. What does mean by introduced supervision S. T is an integer, not an interval, right? What value can T be selected? T should be factors of N?

     - "However, in the past early exit scenarios, the heavy misalignment of path to obtain Y^_i and Y^_{i+1} may cause optimization direction interference intrinsically" I don't get the main point of this sentence.
    - "while the shallow features are used for late classification and early decision both", what do late classification and early decision mean?
    - theorem should be rigorously proved but theorem 2 is just a statement
    - "generalization ability of shallow layers should be enhanced". generalization of deep layers should also be enhanced right?
    - typos: Dyn-Adpater should be Adapter
    - There are two rows for Dyn-XXX in table 1. The author should explain what they are in the caption.


- Some statements in the paper are not clear and confusing.
    - "By freezing the backbone and only updating the adapter to assume the task-related semantic abstraction, we guarantee that the low-level feature in the backbone will not be interfered by loss signal, realizing explicit decoupling of low level feature and high level semantics."  Updating adapters of shallow layers will change low-level features. I don't see how updating adapters can guarantee that the low-level feature in the backbone will not be interfered by loss signal. How can updating adapters fix the problem that early classifiers
force intermediate low-level features to encapsulate high-level semantics and be linearly separable?
    - "Nevertheless, current PETL methods are only capable of tuning the high-level semantic representations and can not directly utilize low-level information in case of the demand of down-stream tasks, thereby undermining their capacity to adapt to diverse datasets." Because of the skip connections in transformer blocks, prediction can also use the features from the shallow layers [1].
    -Because of theorem 1, the authors proposed to use MLP for early stages and light-weight head for later stages. What’s the logic and rationale behind it?
    - For Adaptive Prioritization, does it mean that the lambda needs to adjust manually during the training? The description of the adjustment is very vague, "progressively increase, gradually declines".  How much does it increase/decrease, and how often? The authors need to make this part more clear.


- The increase in the number of parameters is quite significant. 75% for adapter, 59% for LoRAm, and 73% for Repadapter in table 1. In table 5, the number of parameters is doubled.


- Related work. Multiple works have proposed to leverage intermediate features in PETL or transfer learning [2, 3, 4]. The authors should discuss them in the paper.

[1] Revisiting Vision Transformer from the View of Path Ensemble
[2] Visual query tuning: Towards effective usage of intermediate representations for parameter and memory efficient transfer learning
[3] Head2toe: Utilizing intermediate representations for better transfer learning
[4] Lst: Ladder side-tuning for parameter and memory efficient transfer learning

**Questions:**

- Since the number of epochs is fixed, the reported numbers are accuracies at the end of 100 epochs?
- This paper shows that ReAdapter is accepted at ICCV23 but I can't find them in the ICCV accepted paper repository.

---

### Official Review · Reviewer_pscv · 2023-11-05

**Soundness:** 3 good
**Presentation:** 4 excellent
**Contribution:** 2 fair
**Rating:** 5
**Confidence:** 3

**Summary:**

This paper propose a novel parameter-efficient transfer learning technique, dynamic adapter, which motivates from an observation: existing approaches exploit a rigid form of transferring the knowledge from an existing trained model to its downstream tasks might lead to suboptimal performance. They claim this is due to the overly feature compression on the trained task. Their novel approaches exploits a simple yet efficient dynamic mechanism, by freezing the backbone and only train the adapter. More interestingly, their method can be combined with state-of-the-art methods such as LoRA, Adapter, RepAdapter to not only reduce the inference latency while enhancing the performance.

**Strengths:**

This paper solves a timely issue in current transfer learning domain, parametric efficient transfer learning, which is commonly used in many models.

This paper's motivation is clear and solid, that directly transfering the parameters of an given architecture might leads to inferior performance as the original trained features might be over compressed

The solution is relatively simple yet effective, they propose a multi-exit head structure to be jointly optimized in a chain manner, which on the other hand can greatly accelerate the inference speed with a multi-head early stop mechanism.

**Weaknesses:**

Theorem 2 does not really count as a theorem, as there is no clear derivation of mathematical procedure. It is more or less like a conjecture.

This paper plugs their RepAdapter on top of well known method like LoRA, it will be super interesting to see the performance on language models rather than pure toy image datasets.

Given this, I am not sure the practical impact of this work is fully evaluated. I suggest the authors try to provide certain explanation regarding these.

**Questions:**

See above